# Purchase and use of antimicrobials in the hospital sector of Vietnam, a lower middle-income country with an emerging pharmaceuticals market

**Vu Quoc Dat**[1,2,3]*, **Phan Khanh Toan**[1], **H. Rogier van Doorn**[3,4], **C. Louise Thwaites**[3,4], **Behzad Nadjm**[3,5]

**1** Department of Infectious Diseases, Hanoi Medical University, Hanoi, Vietnam, **2** Hanoi Medical University Hospital, Hanoi, Vietnam, **3** Wellcome Africa Asia Programme, Oxford University Clinical Research Unit, Hanoi, Vietnam, **4** Nuffield Department of Clinical Medicine, Centre for Tropical Medicine and Global Health, University of Oxford, Oxford, United Kingdom, **5** MRC The Gambia at the London School of Hygiene & Tropical Medicine, Fajara, The Gambia

* datvq@hmu.edu.vn, quocdat181@yahoo.com

**Data Availability Statement:** All relevant data are within the paper and its Supporting Information files.

## Abstract

### Introduction

Antimicrobial use is associated with emergence of antimicrobial resistance. We report hospital antimicrobial procurement, as a surrogate for consumption in humans, expenditure and prices in public hospitals in Vietnam, a lower middle-income country with a high burden of drug resistant infections.

### Method

Data on antimicrobial procurement were obtained from tender-winning bids from provincial health authorities and public hospitals with detailed bids representing 28.7% (1.68 / 5.85 billion US $) of total hospital medication spend in Vietnam. Antimicrobials were classified using the Anatomical Therapeutic Chemical (ATC) Index and the 2019 WHO Access, Watch, Reserve (AWaRe) groups. Volume was measured in number of Defined Daily Doses (DDD). Antimicrobial prices were presented per DDD.

### Results

Expenditure on systemic antibacterials and antifungals accounted for 28.6% (US $482.6 million/US $1.68 billion) of the total drug bids. 83% of antibacterials (572,698,014 DDDs) by volume (accounting for 45.5% of the antibacterials spend) were domestically supplied. Overall, the most procured antibacterials by DDD were second generation cephalosporins, combinations of penicillins and beta-lactamase inhibitors, and penicillins with extended spectrum. For parenteral antibacterials this was third generation cephalosporins. The average price for antibacterials was US $15.6, US $0.86, US $0.4 and US $11.7 per DDD for Reserve, Watch, Access and non-recommended/unclassified group antibacterials, respectively.

**Funding:** This work was supported by the
Wellcome Trust of Great Britain (106680/Z/14/Z).

**Competing interests:** The authors have declared
that no competing interests exist.

**Abbreviations:** ATC, Anatomical Therapeutic
Chemical; AWaRe, Access, Watch and Reserve;
DDD, Defined Daily Dose; DoH, Department of
Health; LMIC, lower-middle income country; WHO,
World Health Organisation.

## Conclusions

Antimicrobials accounted for a substantial proportion of the funds spent for medication in public hospitals in Vietnam. The pattern of antibacterial consumption was similar to other countries. The high prices of Reserve group and non-recommended/unclassified antibacterials suggests a need for a combination of national pricing and antimicrobial stewardship policies to ensure appropriate accessibility.

## Introduction

Despite concerted international efforts, antimicrobial use continues to rise in both humans and animals. Data on antibacterial sales from 76 countries (including Vietnam) between 2010 and 2015 estimated global antibacterial consumption in humans has increased by 65% over this period, reaching 42 billion defined daily doses (DDDs) every year [1]. Global consumption is forecast to increase by a further 200% between 2015 and 2030 if there are no changes in current practice [1]. The difference in overall antibacterial consumption between the highest and lowest -consuming countries was 3-fold for total use (in DDDs per 1 000 population per day), and up to 16 fold in volume for quinolones and cephalosporins among the (mostly high-income) countries in the Organisation for Economic Co-operation and Development (OECD) [2]. Antibacterial consumption was positively correlated with growth in per capita gross domestic product (GDP) [1] and low- and middle-income countries (LMICs) are consequently responsible for driving the rise in global antibacterial consumption [3]. From 2000 to 2010, Brazil, Russia, India, China and South Africa contributed 30% of global population growth but 76% of the increase in global antibacterial consumption (in number of doses) in the same period [3].

There is a positive correlation between antibacterial consumption and levels of bacterial resistance to antibacterials [4]. In 2017, WHO introduced the AWaRe classification of antibacterials (Access, Watch and Reserve groups) to promote antimicrobial stewardship at local, national and global level, and address the challenge of increased antimicrobial resistance. The 'Access' group includes first and second choice antibacterials for the empirical treatment of common infectious syndromes which should be widely available in all healthcare settings. The 'Watch' group includes antibacterial classes, against which there is a higher resistance potential and are recommended for a limited number of indications. Finally, the 'Reserve' or last resource group includes antibacterials that are recommended for highly specific situations when all alternatives have failed [5]. In October 2019, WHO revised the AWaRe classification to include several antibacterials (mostly second generation cephalosporins) which were not classified in the 2017 version in the 3 existing categories; to classify non evidence-based fixed dose combinations of antibacterials as 'Not recommended' and to link antibacterials with the Anatomical Therapeutic Chemical (ATC) codes and WHO Essential Medicines List [6]. WHO recommends that countries monitor the consumption of the Watch and Reserve antibacterials carefully as part of their AMR strategy [5] and to inform policies which optimize their use in a timely manner [7].

Vietnam is a LMIC with a population of 94.6 million and GDP per capita of US $2,171 [8]. In 2016, health expenditure accounted for 5.7% of GDP, corresponding to annual per capita health expenditure of US $122.8, 45% of which was out of pocket spending [8]. Vietnam has one of the highest rates of antimicrobial drug resistance in Asia. In an antimicrobial resistance surveillance network of 16 hospitals in Vietnam between 2012 and 2013, the proportion of antimicrobial resistance was high among all pathogens isolated from clinical specimens:

penicillin non-susceptible *Streptococcus pneumoniae* (67%, 229/344 isolates), methicillin-resistant *Staphylococcus aureus* (MRSA) (69%, 1098/1580 isolates), third-generation cephalosporin-resistant *Escherichia coli* (56%, 2342/4192 isolates) and *Klebsiella pneumoniae* (66%, 1479/2227 isolates), carbapenem-resistant *Pseudomonas aeruginosa* (33%, 578/1765 isolates) and carbapenem-resistant *Acinetobacter spp.* (70%, 1495/2138 isolates) [9]. However, due to lack of resources for collecting reliable data and maintaining surveillance system, data on antibacterial consumption from LMICs are still limited and of poor quality, especially for countries from Southeast Asia [10]. In addition to quantifying consumption of antimicrobials, understanding relative prices and purchases of antimicrobials is important since their consumption is associated with antimicrobial resistance but a high price of medication may act as a barrier to access, reducing consumption. Our study reports the availability and price of antibacterials and estimates their usage in public hospitals in Vietnam.

## Materials and methods

### Study approach

Data on antimicrobial procurement were obtained from tender-winning bids from 52/63 provincial health authorities and 30 public hospitals across Vietnam for 2018. The process of bidding for contracts to supply medication to public health facilities follows Vietnamese government guidance [11].The current medication procurement in Vietnam is mostly implemented through bidding which uses a decentralised (individual hospitals directly conduct the procurement) or centralised model (at national level by ministry of health or at provincial level by provincial departments of health, DoHs). At provincial level, centralised procurement involves provincial DoHs gathering procurement needs of provincial and districts hospitals under their jurisdiction, calling for, reviewing and accepting bids. Hospitals' estimated requirements for antibacterials are based on consumption in the previous year. Payment is made by the hospitals regardless of whether a decentralised or centralised bid model was used. As part of the procurement regulations, the health facility is expected to ensure the consumption of at least 80% of each medication purchased [11].

The tender-winning bids used for the study comprised 52/63 Provincial Departments of Health, 23 secondary hospitals and 7 primary hospitals (outside the 52 provincial departments) throughout Vietnam. As of December 2017, Vietnam had 13,583 public healthcare facilities, including 1,085 hospitals with 308,400 patient beds, 579 regional clinics and 11,830 medical service units in communes, wards, offices and state- or privately-owned enterprises [12, 13]. In the private sector, there were 231 private hospitals with 16,000 beds (approximately 5% of national hospital beds) in the country by 2019 [14]. The country's public healthcare system is divided into four technical categories: tertiary hospitals (under administrative control of or appointed by the Ministry of Health), secondary hospitals (under the DoHs and catering to and receiving referrals from the province population), primary hospitals (district hospitals under Provincial DoH, catering to and receiving referrals from the district population and commune health stations), and commune health stations or medical service units [15, 16]. Currently there are 75 tertiary hospitals, 491 secondary hospitals, 514 primary hospitals and 5 unclassified hospitals [12].

### Data resources

Data on the price and characteristics of procured antimicrobials in Vietnam were taken from the successful tenders for medicines in 2018 for hospitals and provincial DoHs in Vietnam as published on the website of the Drug Administration of Vietnam which is the Ministry of Health regulatory authority [17]. The bid winning tenders from provincial DoHs may cover all

or only some of the primary and/or secondary hospitals within that province and the data on precisely which hospitals, or the breakdown by type within each bid, were not available. All successful bids with available data were used for this analysis (S1 Fig). The data for each tender included the name of the active ingredient, trademarks, strength, dosage and package, route of administration, registration identification, manufacturer, country of origin of manufacturer, measuring unit, bid quantity, unit price, total value, bidders, suppliers and brand name/ generic name [11]. The list of drugs in our analysis excluded antimicrobial medications which are nationally procured, such as those for the treatment of HIV, influenza, tuberculosis and malaria. The medications procured would be dispensed for inpatients and outpatients in the hospital sector. We described the antimicrobial manufacturers by their country of origin to estimate the market shares between domestic and international manufacturers which may provide some insight into manufacturing capacity.

### Estimation of antimicrobial procurement and the price to the hospitals

Currently, there are no national stewardship programmes defining access to different antimicrobials but individual hospitals may have their own policies on their use. All antimicrobials for systemic use were included in the analysis and classified using the Anatomical Therapeutic Chemical (ATC) Index with Defined Daily Doses (DDDs) 2018 [18]. The DDD is recommended by WHO as a measurement unit of drug consumption [19]. It is the average maintenance dose of a drug per day for a 70 kg adult for its main indication. It provides an estimate and comparison of drug consumption between population groups and is widely used in pharmacoeconomical studies. The DDD for a given drug is assigned by ATC/DDD classification with a unique code and may be different for the routes of administration (oral and parenteral) of the same drug when there is a substantially difference of bioavailability. It is neither defined for topical products nor available for all drug combinations [20]. Antibacterials were further classified by 2019 AWaRe categories [6].

In this analysis, price was defined as a monetary value of an antimicrobial established in successful bids and calculated per DDD. We calculated the total DDD procured for an antimicrobial by multiplying the total procured at each dose-route of administration for this drug by the DDD conversion factor for the corresponding dose-route of administration. The average price per DDD of an antimicrobial drug was calculated by dividing bidding price for that drug by the total number of DDD. The high/low ratio was used to compare the difference between the highest unit price and the lowest price of one DDD of each antimicrobial across all the tenders. All prices were converted from VND to US $ according to the annual average official exchange rates of the World Bank in 2017 (US $1 = 22,370.09 VND) [21]. The average price of antimicrobials per DDD, the share of antimicrobials bidding price in the hospital drug spend and the number of DDD are used to compare in different levels of hospitals. Pareto chart (ABC analysis) was used to examine the consumption of antimicrobials and expenditures for procurement [22, 23]. The ratio of the highest to the lowest price of antimicrobials per DDD (high/low (H/L) ratio) was calculated to report the variation of antimicrobials price [24]. Spearman correlation coefficient was used to assess the association between variation of antimicrobials prices (high/low ratio) and number of manufacturers. Descriptive statistics were performed using Microsoft Excel (Office 365, version 1909, Microsoft Corporation, Redmond, Washington,).

### Results

We included tender-winning results totalling US $1.68 billion from 23 secondary hospitals, 7 primary hospitals and 52 provincial departments of health in Vietnam. This excludes

**Table 1. Expenditure and number of DDD.**

| | | Department of Health (n = 52) | Secondary hospital (n = 23) | Primary hospital (n = 7) | All sites |
|---|---|---|---|---|---|
| J01_antibacterials for systemic use | Total expenditure (%) | US $430,713,755 (29.50%) | US $47,888,882 (22.01%) | US $1,426,924 (18.49%) | US $480,029,561 (28.48%) |
| | Number of DDD (%) | 636,851,337 (93.26%) | 51,468,583 (95.32%) | 4,036,865 (96.62%) | 692,356,785 (93.43%) |
| J02_antimycotics for systemic use | Total expenditure (%) | US $1,750,265 (0.12%) | US $815,819 (0.37%) | US $13,898 (0.18%) | US $2,579,982 (0.15%) |
| | Number of DDD (%) | 2,342,056 (0.34%) | US $219,822 (0.41%) | 20,798 (0.50%) | 2,582,676 (0.35%) |
| P01_antiprotozoals | Total expenditure (%) | US $940,694 (0.06%) | US $17,942 (0.01%) | US $1,577 (0.02%) | US $960,214 (0.06%) |
| | Number of DDD (%) | 9,657,984 (1.41%) | 406,181 (0.75%) | 34,875 (0.83%) | 10,099,040 (1.36%) |
| P02_anthelmintics | Total expenditure (%) | US $436,790 (0.03%) | US $68,622 (0.03%) | US $2,369 (0.03%) | US $507,781 (0.03%) |
| | Number of DDD (%) | 4,735,531 (0.69%) | 590,841 (1.09%) | 17,250 (0.41%) | 5,343,622 (0.72%) |
| Other medications (non-antimicrobial) | Total expenditure (%) | US $1,026,229,647 (70.29%) | US $167,855,286 (77.16%) | US $6,225,722 (80.66%) | US $1,188,532,427 (70.52%) |
| Total | Total espenditure (%) | US $1,460,071,152 (100.00%) | US $217,553,353 (100.00%) | US $7,718,581 (100.00%) | US $1,685,343,086 (100.00%) |
| | Number of antimicrobial DDD (%) | 682,893,636 (100.00%) | 53,994,079 (100.00%) | 4,178,096 (100.00%) | 741,065,810 (100.00%) |

disposable and consumable medical supplies and medical equipment. The estimated total pharmaceutical sales in Vietnam in 2018 was US $ 5.85 billion [25], our analysis (US $1.68 billion) is therefore estimated to represent 28.7% of the funds spent on medication nationally. The overall spending on systemic antibacterials and antifungals accounted for 28.6% (US $482.6 million) of the total funds spent on drugs for the study hospitals (Table 1).

Among antibacterials for systemic use (J01), there were a total of 77 different substances (ATC 5th level) in 23 chemical subgroups (ATC 4th level) procured over all sites. Antibacterials procured according to their AWaRe categories are presented in Fig 1. The Access group, and Watch group antibacterials accounted for 47.2% and 52.4 of procured number of antimicrobial

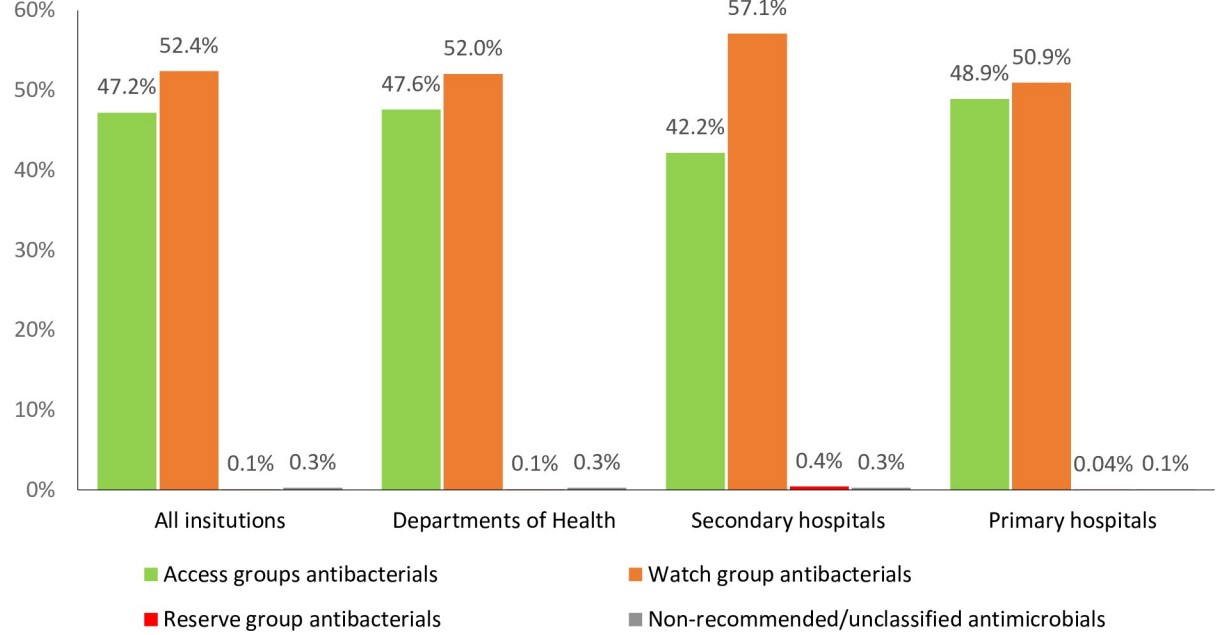

**Fig 1. Proportional antibacterial procurement in DDD (%) by AWaRe classification.**

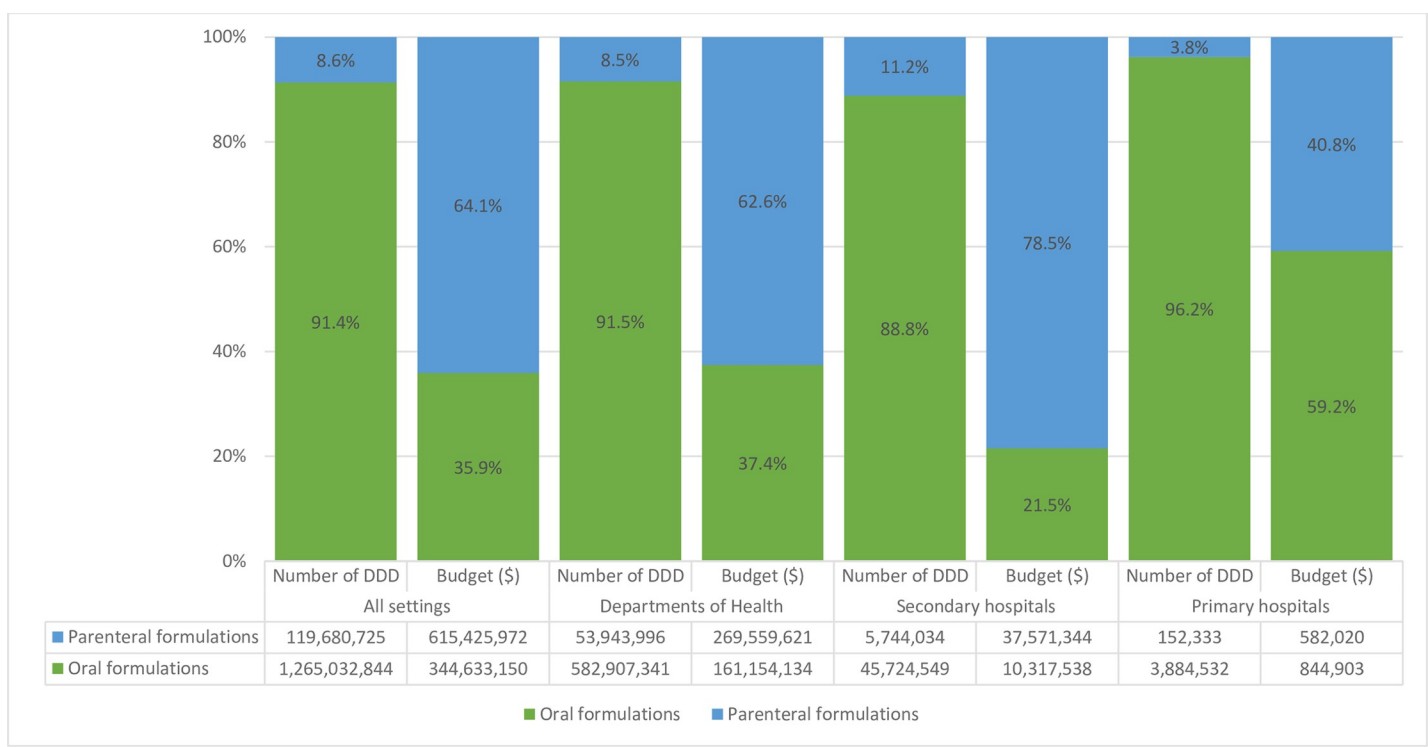

**Fig 2. The proportion of bidding price and number of DDD of antimicrobials for systemic antibacterial (J01) by route of administration in hospitals in Vietnam.**

DDD respectively whilst the Reserve group accounted for 0.1% of antibacterials procured. We identified 4 antibacterials that were unclassified by the 2019 AWaRe classification (ticarcillin with a beta-lactamase inhibitor (J01CR03), nalidixic acid (J01MB02), norfloxacin and tinidazole (J01RA13) and tinidazole (J01XD02)) but remain recommended by national treatment guidelines for specific infections. The proportion of AWaRe non-recommended/unclassified antibacterials were 0.3% of total DDD numbers in all settings. The proportion of Access group, Watch group, Reserve group and non-recommended/unclassified antibacterials provided by domestic manufacturers were 82.2%, 83.3% and 44.3% and 65.7% respectively.

Oral antibacterials accounted for 91.4% of total DDD of antibacterials (J01) across all sites (Fig 2). Parenteral antibacterials represented 11.2% of the procured antibacterial DDDs in secondary hospitals, 3.8% in primary hospitals and 8.5% in bids by DoH. The most common oral antibacterials across all sites were the second generation cephalosporins (J01DC) (cefoxitin, cefamandole, cefmetazole, cefotiam, cefaclor and cefuroxime) (19.8% of total DDD). When stratified by the site of procurement, the most commonly procured oral antibacterials in secondary hospitals were combinations of penicillins and beta-lactamase inhibitors (J01CR) (29%) and in primary and departments of health hospitals were the second generation cephalosporins (J01DC) (21.8% and 20.3% respectively). For parenteral antibacterials, the most common antibacterials were the third generation cephalosporins (J01DD) (29.1%). The details of antibacterial procurement in DDD is shown in Table 2.

The price of antibacterials relative to DDD and bidding cost are shown in Table 3. The second generation cephalosporins (J01DC), combinations of penicillins, including beta lactamase inhibitors (J01CR), penicillins with extended spectrum (J01CA), third generation cephalosporins (J01DD) and fluoroquinolones (J01MA) accounted for 76.6% of all DDD, reaching 65.7%

**Table 2. The proportions of total DDD and expenditure of antibacterials for systemic use (J01).**

| | Department of Health | | Secondary hospitals | | Primary hospitals | | All sites | |
|---|---|---|---|---|---|---|---|---|
| | % DDD | % expenditure | % DDD | % expenditure | % DDD | % expenditure | % DDD | % expenditure |
| J01DC_Second generation cephalosporins | 20.29% | 16.52% | 19.66% | 8.06% | 21.81% | 15.87% | 20.25% | 15.67% |
| J01CR_Combinations of penicillins, incl. beta lactamase inhibitors | 16.31% | 15.77% | 26.33% | 15.22% | 18.35% | 24.10% | 17.06% | 15.74% |
| J01CA_Penicillins with extended spectrum | 15.78% | 2.94% | 10.67% | 0.91% | 11.89% | 2.21% | 15.38% | 2.74% |
| J01DD_Third generation cephalosporins | 12.23% | 23.02% | 19.73% | 23.31% | 8.55% | 24.86% | 12.77% | 23.05% |
| J01MA_Fluoroquinolones | 11.33% | 11.56% | 12.89% | 14.89% | 11.54% | 7.94% | 11.45% | 11.89% |
| J01DB_First generation cephalosporins | 9.00% | 8.16% | 0.80% | 0.57% | 15.00% | 8.52% | 8.43% | 7.40% |
| J01FA_Macrolides | 8.22% | 3.41% | 4.05% | 1.31% | 9.20% | 8.81% | 7.91% | 3.22% |
| J01AA_Tetracyclines | 1.79% | 0.07% | 1.78% | 0.47% | 2.13% | 0.08% | 1.79% | 0.11% |
| J01CE_Beta lactamase sensitive penicillins | 1.59% | 0.08% | 0.07% | 0.00% | 0.39% | 0.03% | 1.47% | 0.07% |
| J01EA_Trimethoprim and derivatives | 0.97% | 0.18% | 0.22% | 0.01% | 0.11% | 0.01% | 0.91% | 0.16% |
| J01GB_Other aminoglycosides | 0.86% | 1.63% | 1.45% | 1.27% | 0.72% | 0.84% | 0.90% | 1.59% |
| J01XD_Imidazole derivatives | 0.40% | 1.29% | 0.48% | 0.81% | 0.05% | 0.22% | 0.40% | 1.24% |
| J01CF_Beta lactamase resistant penicillins | 0.39% | 0.90% | 0.18% | 0.29% | 0.05% | 0.13% | 0.37% | 0.83% |
| J01FF_Lincosamides | 0.22% | 0.74% | 0.22% | 0.93% | 0.00% | 0.00% | 0.22% | 0.76% |
| J01DH_Carbapenems | 0.18% | 9.18% | 0.50% | 18.74% | 0.04% | 1.27% | 0.20% | 10.12% |
| J01DE_Fourth generation cephalosporins | 0.15% | 2.25% | 0.29% | 3.43% | 0.05% | 2.04% | 0.16% | 2.37% |
| J01MB_Other quinolones | 0.14% | 0.05% | 0.01% | 0.00% | 0.04% | 0.04% | 0.13% | 0.05% |
| J01XX_Other antibacterials | 0.05% | 0.69% | 0.37% | 3.09% | 0.04% | 0.76% | 0.08% | 0.93% |
| J01XA_Glycopeptide antibacterials | 0.05% | 0.69% | 0.23% | 2.78% | 0.03% | 1.05% | 0.06% | 0.90% |
| J01BA_Amphenicols | 0.03% | 0.03% | 0.00% | 0.00% | 0.01% | 0.01% | 0.03% | 0.02% |
| J01XB_Polymyxins | 0.01% | 0.84% | 0.07% | 3.90% | 0.01% | 1.19% | 0.02% | 1.15% |

The cells were colorized with red-yellow-green color scale by column. The highest values in a column were red, the average values were yellow, and the lowest values were green. DDD = Defined Daily Dose.

of the total spend across all sites. However, carbapenems (J01DH) only accounted for 0.3% of antibacterial use but 10.2% of the total spent on antibacterials (Fig 3).

By AWaRe categories, the average price per DDD of Reserve group antibacterials was the highest (US $15.63 per DDD), followed by the Watch group antibacterials (US $0.86 per DDD), and Access group antibacterial (US $0.4 per DDD). The average price of non-recommended/unclassified antibacterials was 11.7 per DDD. We present the price of antimicrobials (ATC 5th level) in Table 3. The three most expensive antimicrobials were caspofungin (J02AX04) (US $284.5 per DDD), doripenem (J01DH04) (US $85.3 per DDD) and tigecycline (J01AA12) (US $65.4 per DDD). There is a large variability in antimicrobial price per DDD range which is represented as a ratio of the highest to lowest price of antimicrobial per DDD (H/L ratio) from very high (H/L ratio up to 82 for oral formulation or 40.26 for parenteral formulation) or no discrepancy (H/L = 0) for the branded forms (Table 3). Twenty-seven of thirty-eight (71.3%) oral forms and 10/55 (18.2%) parenteral forms of antimicrobials had H/L ratios above 10.

Whilst almost all (59/77, 76.6%) antibacterials for systemic use (ATC 5th level, chemical substance) were procured from both domestic and international manufacturers, 18 were procured from either international or domestic manufactures (Fig 4 and Table 4). 82.7% of antibacterials were supplied by domestic producers (67 companies, supplied 572,698,014 DDDs) whilst 212 international manufactures from 35 countries supplied the remainder (119,658,771 DDD). Antibacterials supplied by international manufacturers accounted for 54.5% of the

**Table 3. Prices of antimicrobials.**

| Antimicrobials | Number of samples | No. Mfr. | Average price per DDD | High/low ratio | Number of samples | No. Mfr. | Average price per DDD | High/low ratio | Number of samples | No. Mfr. | Average price per DDD | High/low ratio |
|---|---|---|---|---|---|---|---|---|---|---|---|---|
| | | **All** | | | | **Oral formulation** | | | | **Parenteral formulation** | | |
| J02AX04_caspofungin | 7 | 1 | 284.53 | 1.10 | - | - | | | 7 | 1 | 284.53 | 1.10 |
| J01DH04_doripenem | 10 | 4 | 85.31 | 1.56 | - | - | | | 10 | 4 | 85.31 | 1.56 |
| J01AA12_tigecycline | 10 | 1 | 65.36 | 1.00 | - | - | | | 10 | 1 | 65.36 | 1.00 |
| J01XB01_colistin | 84 | 5 | 49.98 | 2.18 | - | - | | | 84 | 5 | 49.98 | 2.18 |
| J01CR03_ticarcillin and beta lactamase inhibitor | 64 | 2 | 40.87 | 3.56 | - | - | | | 64 | 2 | 40.87 | 3.56 |
| J01DH51_imipenem and cilastatin | 183 | 21 | 35.71 | 6.33 | - | - | | | 183 | 21 | 35.71 | 6.33 |
| J01CA12_piperacillin | 15 | 2 | 34.88 | 1.80 | - | - | | | 15 | 2 | 34.88 | 1.80 |
| J01DH02_meropenem | 269 | 21 | 30.98 | 12.90 | - | - | | | 269 | 21 | 30.98 | 12.90 |
| J01DC01_cefoxitin | 145 | 13 | 26.57 | 8.32 | - | - | | | 145 | 13 | 26.57 | 8.32 |
| J01DH03_ertapenem | 15 | 1 | 24.69 | 1.00 | - | - | | | 15 | 1 | 24.69 | 1.00 |
| J01CR05_piperacillin and beta lactamase inhibitor | 151 | 14 | 18.51 | 3.90 | - | - | | | 151 | 14 | 18.51 | 3.90 |
| J01DE02_cefpirome | 72 | 9 | 18.00 | 3.59 | - | - | | | 72 | 9 | 18.00 | 3.59 |
| J01XA02_teicoplanin | 69 | 5 | 17.75 | 2.23 | - | - | | | 69 | 5 | 17.75 | 2.23 |
| J01DD62_cefoperazone and beta lactamase inhibitor | 196 | 21 | 16.56 | 27.33 | - | - | | | 196 | 21 | 16.56 | 27.33 |
| J01DC03_cefamandole | 66 | 8 | 15.88 | 4.35 | - | - | | | 66 | 8 | 15.88 | 4.35 |
| J01DB03_cefalotin | 70 | 4 | 14.55 | 1.77 | - | - | | | 70 | 4 | 14.55 | 1.77 |
| J01CR01p_ampicillin and beta lactamase inhibitor | 159 | 12 | 11.55 | 6.06 | - | - | | | 159 | 12 | 11.55 | 6.06 |
| J02AA01_amphotericin B | 23 | 2 | 11.30 | 12.10 | - | - | | | 23 | 2 | 11.30 | 12.10 |
| J01DC09_cefmetazole | 75 | 8 | 10.06 | 3.06 | - | - | | | 75 | 8 | 10.06 | 3.06 |
| J01XX01_fosfomycin | 154 | 8 | 9.12 | 2.81 | 44 | 2 | 5.13 | 1.52 | 110 | 6 | 10.31 | 2.62 |
| J01DD07_ceftizoxime | 108 | 15 | 7.95 | 6.18 | - | - | | | 108 | 15 | 7.95 | 6.18 |
| J01DD12_cefoperazone | 134 | 15 | 7.83 | 12.68 | - | - | | | 134 | 15 | 7.83 | 12.68 |
| J01FA10p_azithromycin | 21 | 6 | 7.62 | 6.90 | - | - | | | 21 | 6 | 7.62 | 6.90 |
| J01XA01_vancomycin | 166 | 15 | 7.57 | 2.57 | - | - | | | 166 | 15 | 7.57 | 2.57 |
| J01DE01_cefepime | 157 | 16 | 7.18 | 19.53 | - | - | | | 157 | 16 | 7.18 | 19.53 |
| J01MA14_moxifloxacin | 225 | 16 | 6.93 | 37.69 | 61 | 7 | 0.92 | 5.38 | 164 | 9 | 12.09 | 3.06 |
| J01CR02p_amoxicillin and beta lactamase inhibitor | 149 | 12 | 6.58 | 6.52 | - | - | | | 149 | 12 | 6.58 | 6.52 |
| J01DB12_ceftezole | 67 | 10 | 6.07 | 2.90 | - | - | | | 67 | 10 | 6.07 | 2.90 |
| J01DD02_ceftazidime | 305 | 26 | 5.07 | 11.56 | - | - | | | 305 | 26 | 5.07 | 11.56 |
| J01GB07_netilmicin | 90 | 9 | 4.99 | 3.43 | - | - | | | 90 | 9 | 4.99 | 3.43 |
| J01XX08_linezolid | 32 | 8 | 4.67 | 40.45 | 17 | 4 | 1.70 | 1.57 | 15 | 4 | 24.76 | 3.68 |
| J01XD02_tinidazole | 25 | 4 | 4.61 | 2.98 | - | - | | | 25 | 4 | 4.61 | 2.98 |
| J01DD04_ceftriaxone | 263 | 28 | 3.51 | 40.26 | - | - | | | 263 | 28 | 3.51 | 40.26 |
| J01DC07_cefotiam | 40 | 10 | 3.05 | 4.00 | - | - | | | 40 | 10 | 3.05 | 4.00 |
| J01DD01_cefotaxime | 285 | 29 | 2.77 | 12.18 | - | - | | | 285 | 29 | 2.77 | 12.18 |
| J01DB04_cefazolin | 92 | 9 | 2.46 | 4.18 | - | - | | | 92 | 9 | 2.46 | 4.18 |
| J01FF01_clindamycin | 189 | 13 | 2.40 | 39.58 | 66 | 10 | 0.43 | 9.11 | 123 | 3 | 5.26 | 3.93 |
| J01DD14_ceftibuten | 15 | 2 | 2.37 | 2.46 | 15 | 2 | 2.37 | 2.46 | 0 | - | | |
| J01GB06_amikacin | 202 | 12 | 2.25 | 8.33 | - | - | | | 202 | 12 | 2.25 | 8.33 |
| J01CF02_cloxacillin | 107 | 9 | 2.05 | 34.20 | 41 | 3 | 0.95 | 5.18 | 66 | 6 | 4.76 | 4.94 |

*(Continued)*

**Table 3.** (Continued)

| Antimicrobials | All | | | | Oral formulation | | | | Parenteral formulation | | | |
|---|---|---|---|---|---|---|---|---|---|---|---|---|
| | Number of samples | No. Mfr. | Average price per DDD | High/low ratio | Number of samples | No. Mfr. | Average price per DDD | High/low ratio | Number of samples | No. Mfr. | Average price per DDD | High/low ratio |
| J01XD01_metronidazole | 160 | 13 | 1.93 | 4.52 | - | - | | | 160 | 13 | 1.93 | 4.52 |
| J01GB01_tobramycin | 99 | 16 | 1.26 | 8.75 | - | - | | | 99 | 16 | 1.26 | 8.75 |
| J02AC02_itraconazole | 148 | 11 | 1.12 | 308.00 | 139 | 10 | 0.71 | 6.33 | 9 | 1 | 82.61 | 1.00 |
| J01CF04_oxacillin | 81 | 6 | 0.85 | 11.50 | 27 | 3 | 0.59 | 2.50 | 54 | 3 | 1.67 | 3.14 |
| J01MA12_levofloxacin | 587 | 39 | 0.81 | 495.43 | 233 | 20 | 0.27 | 72.13 | 354 | 19 | 5.07 | 12.04 |
| J01DD13_cefpodoxime | 230 | 27 | 0.76 | 27.03 | 230 | 27 | 0.76 | 27.03 | - | - | | |
| J01DC04_cefaclor | 372 | 21 | 0.64 | 65.52 | 372 | 21 | 0.64 | 65.52 | - | - | | |
| J01DD15_cefdinir | 174 | 19 | 0.64 | 29.79 | 174 | 19 | 0.64 | 29.79 | - | - | | |
| J01CA01_ampicillin | 85 | 6 | 0.63 | 7.10 | 1 | 1 | 0.12 | 1.00 | 84 | 5 | 0.63 | 2.88 |
| J01MA06_norfloxacin | 20 | 6 | 0.61 | 18.73 | 20 | 6 | 0.61 | 18.73 | - | - | | |
| J01CE01_benzylpenicillin | 41 | 4 | 0.61 | 5.40 | - | - | | | 41 | 4 | 0.61 | 5.40 |
| J02AC01_fluconazole | 173 | 18 | 0.60 | 142.46 | 159 | 16 | 0.46 | 80.30 | 14 | 2 | 12.02 | 1.47 |
| J01MA03_pefloxacin | 34 | 5 | 0.58 | 12.50 | 12 | 3 | 0.14 | 1.71 | 22 | 2 | 1.10 | 1.27 |
| J01MA02_ciprofloxacin | 518 | 39 | 0.56 | 1899.69 | 222 | 24 | 0.08 | 42.81 | 296 | 15 | 8.85 | 31.10 |
| J01BA01_chloramphenicol | 43 | 9 | 0.52 | 5.25 | 24 | 8 | 0.46 | 2.02 | 19 | 1 | 1.34 | 1.14 |
| J01FA02_spiramycin | 194 | 17 | 0.45 | 9.07 | 194 | 17 | 0.45 | 9.07 | - | - | | |
| J01DB09_cefradine | 96 | 8 | 0.43 | 12.76 | 78 | 5 | 0.42 | 12.76 | 18 | 3 | 0.74 | 3.62 |
| J01CR02o_amoxicillin and beta lactamase inhibitor | 965 | 33 | 0.38 | 23.70 | 965 | 33 | 0.38 | 23.70 | - | - | | |
| J01DB05_cefadroxil | 254 | 24 | 0.37 | 24.89 | 254 | 24 | 0.37 | 24.89 | - | - | | |
| J01DD08_cefixime | 550 | 31 | 0.34 | 52.05 | 550 | 31 | 0.34 | 52.05 | - | - | | |
| J01DB01_cefalexin | 304 | 20 | 0.31 | 16.47 | 304 | 20 | 0.31 | 16.47 | - | - | | |
| J01MA01_ofloxacin | 124 | 18 | 0.27 | 677.97 | 97 | 14 | 0.04 | 10.17 | 27 | 4 | 8.42 | 6.53 |
| J01FA09_clarithromycin | 356 | 23 | 0.27 | 19.19 | 356 | 23 | 0.27 | 19.19 | - | - | | |
| J01FA10o_azithromycin | 334 | 27 | 0.26 | 38.66 | 334 | 27 | 0.26 | 38.66 | - | - | | |
| J01DC02_cefuroxime | 876 | 39 | 0.25 | 28.92 | 627 | 28 | 0.21 | 28.92 | 249 | 11 | 0.58 | 9.54 |
| J01MB02_nalidixic acid | 50 | 5 | 0.25 | 1.83 | 50 | 5 | 0.25 | 1.83 | - | - | | |
| J01FA01_erythromycin | 157 | 10 | 0.20 | 82.00 | 157 | 10 | 0.20 | 82.00 | - | - | | |
| J01FF02_lincomycin | 2 | 1 | 0.17 | 1.67 | 1 | 1 | 0.16 | 1.00 | 1 | 1 | 0.27 | 1.00 |
| J01GA01_streptomycin | 1 | 1 | 0.16 | 1.00 | - | - | | | 1 | 1 | 0.16 | 1.00 |
| J01GB03_gentamicin | 88 | 9 | 0.15 | 2.24 | - | - | | | 88 | 9 | 0.15 | 2.24 |
| J01EA01_trimethoprim | 165 | 23 | 0.12 | 70.00 | 165 | 23 | 0.12 | 70.00 | - | - | | |
| J01FA06_roxithromycin | 87 | 14 | 0.11 | 17.14 | 87 | 14 | 0.11 | 17.14 | - | - | | |
| J01CA04_amoxicillin | 413 | 21 | 0.11 | 26.52 | 413 | 21 | 0.11 | 26.52 | - | - | | |
| J01AA07_tetracycline | 22 | 9 | 0.04 | 1.76 | 22 | 9 | 0.04 | 1.76 | - | - | | |
| J02AB02_ketoconazole | 3 | 2 | 0.03 | 1.19 | 3 | 2 | 0.03 | 1.19 | - | - | | |
| J01CE10_benzathine phenoxymethylpenicillin | 88 | 7 | 0.03 | 2.74 | 88 | 7 | 0.03 | 2.74 | - | - | | |
| J01AA02_doxycycline | 83 | 11 | 0.01 | 6.25 | 83 | 11 | 0.01 | 6.25 | - | - | | |

No. Mfr.: number of manufactures; DDD: Defined daily dose.

Note: Number of samples per drug represents the total number of brands at different strengths procured by all bidders. Some brands may have different strengths and the price per DDD may vary for the same drug by the same manufacturer. For example, a 10 mL vial of caspofungin contains either 50 mg or 70 mg, therefore the cost per DDD will be different between 2 strengths. Additionally, the same strengths may have different prices in different provinces.

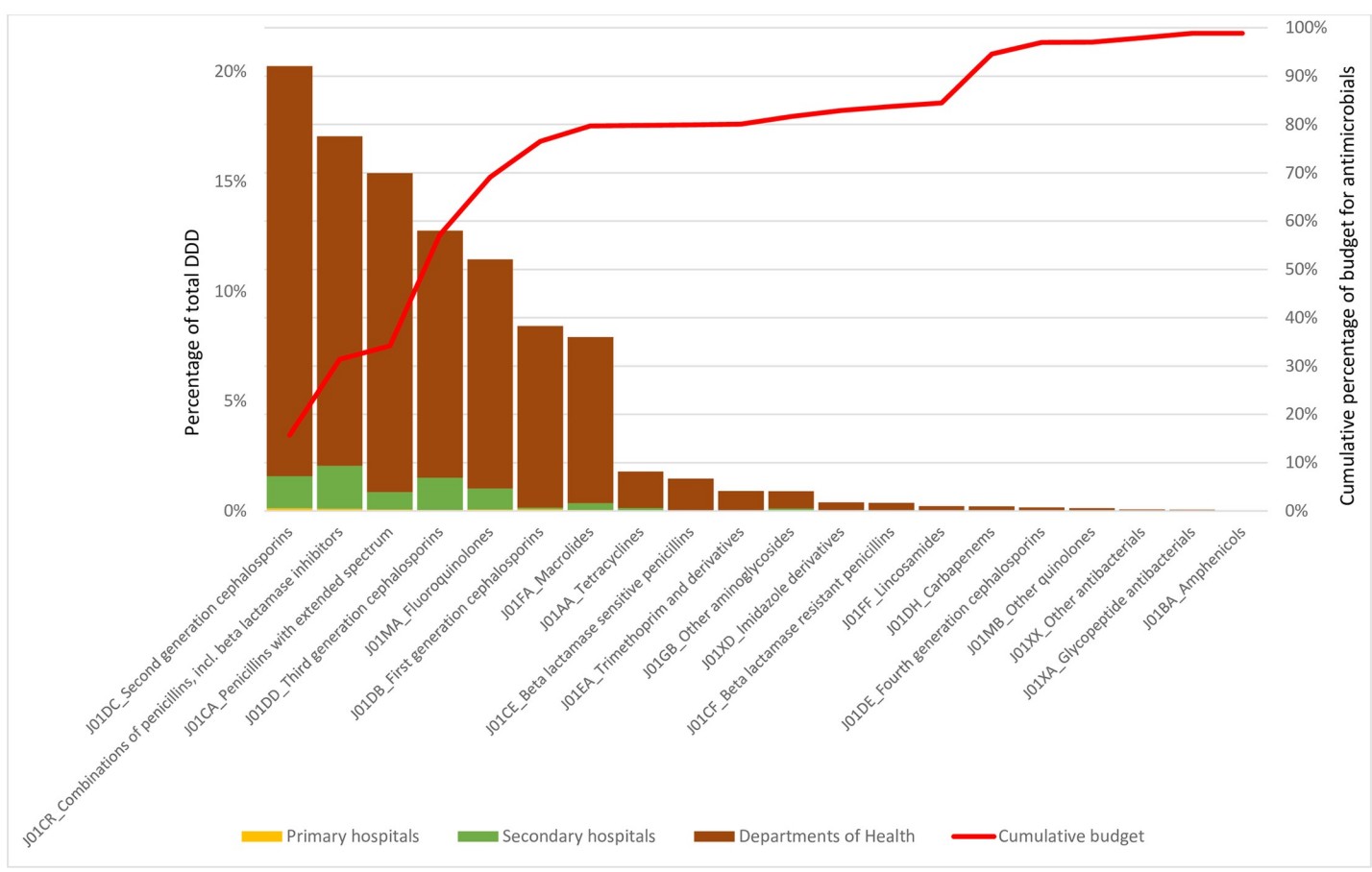

**Fig 3. ABC analysis of the quantities of procured antibacterials (in DDD) and the price.**

total spent (US $261,754,116) and 17.3% of the total DDDs procured. Among 35 countries with pharmaceutical companies sharing the antibacterial market in Vietnam, India and Cyprus contributed the highest proportions of total DDDs, together supplying 8.4% of total antibacterial (58,142,107 DDDs) in all sites, corresponding to 48.6% of total foreign antibacterials (J01) procured (28.6% from India and 20.1% from Cyprus).

## Discussion

Our study represents the first effort to describe the use and price of antimicrobials in healthcare facilities in Vietnam, a country with a high burden of drug resistant infections. In a previous study of antibacterial sales in 76 countries between 2000 and 2015, Vietnam ranked 11th in antibacterial consumption per capita with 32 DDDs per 1,000 inhabitants per day [1]. However, the use of antimicrobials by ATC index and by the route of administration in public hospitals has not been published. Additionally, we found that the proportion of total medication expenditure represented by antibacterials was high—28.5% of the total expenditure for medications across study sites.

The amendment of 2019 AWaRe classification has overcome some limitations of the previous version, especially failures to classify many of the most commonly used drugs. Using the 2017 AWaRe classification, 25.8% of DDD (178.658.638 DDD) procured in our dataset were unclassified, among which 78.5% were 2nd generation cephalosporins (J01DC), 13.4% were

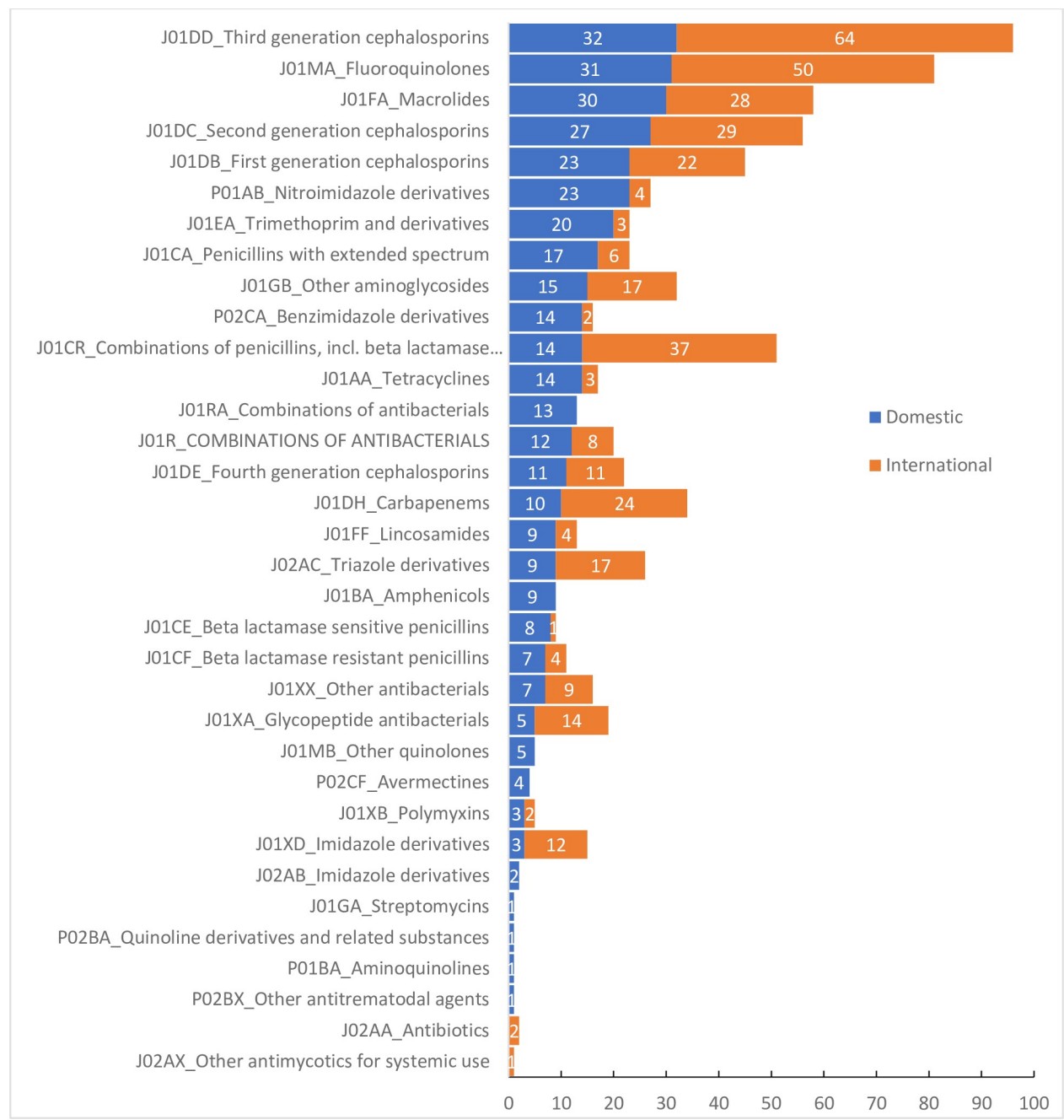

**Fig 4. Number of antimicrobial manufacturers in Vietnam.**

1st generation cephalosporins (J01DB), 5.4% were macrolides (J01FA) and 2.7% were other substances. Similarly, a large proportion of unclassified antibacterials by the 2017 AWaRe system was also reported in other countries, for example 60.3% in a survey of prescriptions among hospitalised children in 56 countries in 2015 [26]. These limitations of the AWaRe classification system were acknowledged by the WHO Essential Medicines List Working Group as requiring further revision [27]. However, the 2019 amendment enables us to re-classify these 'other' antibacterials in our dataset as belonging to the Access group (14.3% of DDD number

**Table 4. Source of manufacturers for selected antibacterials.**

| Antibacterials procured from only domestics manufactures | norfloxacin and tinidazole (J01RA13) |
| --- | --- |
| | lomefloxacin (J01MA07) |
| | streptomycin (J01GA01) |
| | lincomycin (J01FF02) |
| | ciprofloxacin and tinidazole (J01RA11) |
| | ceftibuten (J01DD14) |
| | tetracycline (J01AA07) |
| | pefloxacin (J01MA03) |
| | benzylpenicillin (J01CE01) |
| | chloramphenicol (J01BA01) |
| | nalidixic acid (J01MB02) |
| | ticarcillin and beta lactamase inhibitor (J01CR03) |
| | oxacillin (J01CF04) |
| | gentamicin (J01GB03) |
| | spiramycin and metronidazole (J01RA04) |
| Antibacterial procured from only international manufactures | tigecycline (J01AA12) |
| | ertapenem (J01DH03) |
| | parenteral azithromycin (J01FA10) |

of unclassified antibacterial), Watch group (85% of DDD number of unclassified antibacterial) and non-recommended/unclassified group (0.72% of DDD number of unclassified antibacterial).

We found that cephalosporins were the most commonly prescribed class of antibiotics in Vietnam. In healthcare sectors in Europe, studies report that the most common antibacterials were beta-lactams, penicillins (J01C) [28]. The difference in prescribing patterns may be due to differences in resistance or knowledge. Numerous studies have reported increased resistance in Vietnam compared to Europe [9, 28–30] and in China, where the resistant levels are similar to Vietnam, 3[rd]-generation cephalosporins were the most consumed antibacterial in hospitals [31]. In a systematic review of studies published between 1993 and 2013 about antimicrobial prescription in China (n = 67) and Vietnam (n = 29), the most important factor influencing irrational prescription in Vietnam was lack of knowledge and effective control and regulation mechanisms for drugs use, whilst in China it was financial incentive and lack of knowledge [32]. In the first surveillance report by WHO on antibacterial consumption in 65 countries during the period of 2016–2018, the Philippines was the only country from Southeast Asia providing data on national consumption of antimicrobials for community and hospital use using the IQVIA and import database [10]. In the Philippines, the most frequently consumed antibacterials were tetracyclines and penicillins with each contributing 30% of total consumption of antibacterials (in DDD per 1000 inhabitants per day).

This study provides important data concerning prices of antimicrobials used in Vietnam. A small number of antimicrobials accounted for a disproportionately large high expenditure. Carbapenems were only a small number of antimicrobials prescribed according to DDD, but a significant proportion of the total spend. Data on the national expenditure on antimicrobials is often limited and there are few data from other countries for comparison but it is likely that the situation of last resort antibacterials is similar in other setting. For example, in the US in 2015, the antibacterial expenditures were largest for daptomycin (1.3 billion or 14.7% of total expenditure), and although no data are provided on daptomycin usage, it is likely to be relatively infrequently used.

With a total of 290 manufacturers sharing the antimicrobial market in Vietnam, currently, there are numerous manufacturers with different forms and formulation of antimicrobials on the Vietnamese market. For example, there were eight US Food and Drug Administration–approved manufacturers for Cefuroxime 250 mg tablet in US [33] whilst there were 20 manufacturers approved by the Vietnamese government for the same formulation and strength in Vietnam. This large number of domestic and international manufacturers likely leads to increased complexity in ensuring the quality of medications and regulating their manufacture and distribution. Additionally, for the most common class of antibacterials (second generation cephalosporins), we found that there were 6 drugs used in Vietnam, comparatively high compared to other countries. For example there are only 2 second generation cephalosporins licensed in the UK [34]. The larger number of drugs used in Vietnam may lead to difficulties for clinicians and pharmacists in gaining familiarity with individual drugs and also increases the burden on the government to ensure the quality of the additional number of preparations.

This study has shown high variability in antimicrobial prices. Measurement of price and availability of antimicrobials is essential to inform policies about accessibility and affordability to the population. In a survey of the prices, availability, and affordability of 42 core medicines (including eight antibacterial substances) in 5 provinces in Vietnam in 2005, the prices of innovator drugs and the lowest priced generic drugs were 47 times and 11 times higher than the international reference prices (MSH), respectively [35]. The medicine prices in the public sectors were higher than in private sectors [35]. The reasons for this are unclear. In a qualitative study on the price of medication in Vietnam in 2008, the higher prices in public hospitals were suggested to be related to the high bed occupancy rates resulting in a reduced need to attract more patients through competitive pricing, or prices inflated up to 60% by commissions to prescribers and hospital pharmaceutical departments [36]. However, others have shown financial incentives may be not an important factor influencing doctor's prescribing decision in Vietnam [32]. The International Medical Products Price Guide by Management Sciences for Health (MSH) is recommended as a most useful reference for medicine prices [37] and the 2015 version is the latest [24]. Among antibacterials in the Reserve group, the high/low ratio of cefepime according to MSH 2015 (4.39) was much lower than the one in this study (19.53) [24]. However, no data for other antibacterials in the Reserve group were available for cross reference.

The pharmaceutical market in Vietnam is import-reliant and was estimated to reach US $5.2 billion in 2017 [25]. Ninety percent of the country's medication expenditure was on imported medicines [38]. The European Union was the most important pharmaceutical manufacturer providing medications to Vietnam with a value of US $1.1 billion or 51% of Vietnam's total pharmaceutical imports in 2014, in which France, Germany, Italy shares US $579 million or 73% of total pharmaceutical imports from the EU [39]. We confirmed that European countries were the leading exporters to Vietnam in term of spending on antibacterials whilst Asian countries (mainly India) accounted for the largest quantities. However, the majority of antimicrobial consumption was met by domestic manufacturers. This increase in national production is part of strategic plan for developing the Vietnam pharmaceutical industry by 2020 as the government set objectives to produce 80% of total annual medication consumption in the country [40].

Our study has some limitations. Firstly, we were unable to obtain data from all healthcare institutions in Vietnam due to bids being unpublished or published elsewhere, e.g. an institutional website. Based on the previous estimation of national drug expenditure (US $ 5.85 billion), we estimated our data represents at least 28.7% but this may be an underestimate because our dataset excluded nationally procured medications (antiretrovirals, anti-tuberculosis drug and anti-malarials). Secondly, we excluded the available tender-winning results of US

$ 776,271,465 (equal to 13.4% of estimated national drug expenditure) from 47/75 tertiary hospitals because most of them were highly specialised hospitals and their pattern of antimicrobials use was driven by the specialities and different from general hospitals. This may lead to an under-representation of tertiary hospitals in our analysis and a corresponding underestimation of the contribution of Reserve group antibacterials. Furthermore the data used in the study may not be complete because hospitals may have extra calls for bids or not use all of the antibacterials purchased from these bids. However, the successful bids were based on the previous year's actual (total) consumption and the institutions were required to use at least 80% of antimicrobials purchased in these bids, therefore our estimate is likely to be a reasonable estimate of use. Our data on antimicrobial usage in hospitals may be biased and underestimate exact usage because medications for hospitalised patients can also be purchased directly by patients, especially in primary and secondary hospitals, partly due to the fact that many Watch and Reserve antibacterials are not reimbursed by the national insurance programme and are consequently not available in hospital pharmacies [41]. We have no data to estimate this, but we consider this proportion to be small. In a prospective study of 892 hospitalised trauma patients admitted to a secondary hospital in Vietnam in 2010, the total medical care out-of-pocket cost paid by patients was US $270.6, 23% of this relating to drugs [42]. However, the out-of-pocket cost for antimicrobials in hospitalised patients was not specified. Additionally, there is some uncertainty about the number of hospitals included in this study as the bid by DoH covered the tenders of different public hospitals in the province and the number and nature of hospitals joining these provincial bids was unidentifiable. As electronic health records are currently being implemented in Vietnamese hospitals, it may be possible to obtain more accurate data in future, and to cross reference purchased drugs with prescribed drugs.

A final limitation of this study is that we did not include private sector consumption. As the Vietnamese Government launched a strategy to increase the share of private hospital beds to 20% by 2020 [43], the growth of the private healthcare system may contribute significantly to consumption of antimicrobials.

## Conclusions

Antimicrobials accounted for one third of the total spent on medications in study hospitals in Vietnam. The pattern of antibacterial consumption by AWaRe categories was similar to other countries. However, given the relatively high proportion of antimicrobial drug resistance in Vietnam and that although stewardship is important in reducing it, there is necessity for access in certain circumstances to AWARE reserve antibiotics. The optimal approach would be strict stewardship combined with price control, to allow access when needed and prevent access when not needed.

## Supporting information

**S1 Table. List of available antimicrobials.**
(DOCX)

**S1 Fig.**
(JPG)

## Acknowledgments

We would like to thank Hugo C. Turner (Oxford University Clinical Research Unit) for his review of this manuscript and Trinh Manh Hung (Oxford University Clinical Research Unit) for health economic advice.

## Author Contributions

**Conceptualization:** Vu Quoc Dat, H. Rogier van Doorn, C. Louise Thwaites, Behzad Nadjm.

**Formal analysis:** Vu Quoc Dat, Phan Khanh Toan, C. Louise Thwaites, Behzad Nadjm.

**Methodology:** Vu Quoc Dat, H. Rogier van Doorn.

**Project administration:** H. Rogier van Doorn.

**Supervision:** H. Rogier van Doorn, C. Louise Thwaites, Behzad Nadjm.

**Validation:** Vu Quoc Dat, Phan Khanh Toan, H. Rogier van Doorn, C. Louise Thwaites, Behzad Nadjm.

**Visualization:** Vu Quoc Dat.

**Writing – original draft:** Vu Quoc Dat, Phan Khanh Toan, H. Rogier van Doorn, C. Louise Thwaites, Behzad Nadjm.

**Writing – review & editing:** Vu Quoc Dat, Phan Khanh Toan, H. Rogier van Doorn, C. Louise Thwaites, Behzad Nadjm.

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
