## [Decision Letter · Decision Letter 0]

23 Jul 2020

PONE-D-20-18659

Price and use of antimicrobials in the hospital sector of Vietnam, a lower middle-income country with an emerging pharmaceuticals market

PLOS ONE

Dear Dr. Dat,

Thank you for submitting your manuscript to PLOS ONE. After careful consideration, we feel that it has merit but does not fully meet PLOS ONE’s publication criteria as it currently stands. Therefore, we invite you to submit a revised version of the manuscript that addresses the points raised during the review process.<please by="" manuscript="" revised="" submit="" your="">

Please include the following items when submitting your revised manuscript:</please>

We look forward to receiving your revised manuscript.

Kind regards,

Khin Thet Wai, MBBS, MPH, MA (Population & Family Planning Resear

Academic Editor

PLOS ONE

Additional Editor Comments:

English language correction is deemed necessary.

To carry out an extensive revision for clarity and improvement in scientific integrity.

Reviewers' comments:

Reviewer's Responses to Questions

**Comments to the Author**

1. Is the manuscript technically sound, and do the data support the conclusions?

Reviewer #1: Partly

Reviewer #2: Yes

Reviewer #3: Partly

2. Has the statistical analysis been performed appropriately and rigorously? 

Reviewer #1: Yes

Reviewer #2: Yes

Reviewer #3: N/A

3. Have the authors made all data underlying the findings in their manuscript fully available?

Reviewer #1: Yes

Reviewer #2: No

Reviewer #3: No

4. Is the manuscript presented in an intelligible fashion and written in standard English?

Reviewer #1: Yes

Reviewer #2: Yes

Reviewer #3: No

5. Review Comments to the Author

Reviewer #1: This is an important paper, as there is data on consumption of antibiotics is still relatively scarce, and this is rarely linked to price. The fact that pharmacy and tertiary hospital data means that this is a partial picture, but this paper nevertheless provides a useful contribution to an important issue.

The fact that antibiotics are more than a quarter of the total medicines budget for Vietnam is an important finding. This level of investment is sometimes surprising to policy makers and should encourage them to take antibiotic policies much more seriously. The fact that the proportionate expenditure on reserve antibiotics that are imported, is so high is another policy relevant finding. This study highlights an area where implementation of good public health and prescribing policies could save money and support a shift towards more domestically produced medicines.

The conclusions are slightly surprising. The study has shown significant use and expenditure on Reserve antibiotics and that are not recommended by WHO. The authors conclusion is that there should be price controls on these groups. It is not clear why, as currently financial incentives align with good public health policy.

The case could be made much more strongly that good stewardship programmes, that might shift consumption towards access groups are likely to result in significant cost savings, and support consumption of domestically produced medicines . This argument would increase the relevance of the study to policy makers and the broader policy debates on antibiotic stewardship, and might stimulate investement in stewardship programmes

Most readers won’t be familiar with the Vietnamese health financing system and in particular whether there are financial incentives for practitioners or to hospitals to prescribe antibiotics. It would be helpful if the authors explained whether reimbursement is linked to either the costs of volumes of drugs that are prescribed (as happens in some Asian countries)

Inclusion of antifungals is helpful, and the fact that the consumption in an emerging economy (with relatively low HIV) is relatively low is useful information.

Given that the sample did not include many of the tertiary hospitals total consumption of these products may be even higher. Strengthening stewardship programmes to bring prescribing pr

IN the introduction paragraph the authors should note that data quality on consumption patterns is still relatively poor, and so any rankings are questionable .. The statement that low and middle income countries are disproportionately responsible for the global growth in antibiotics is disingenuous, given that the burden of disease is higher in many of these countries, and their consumption of antibiotics was at a relatively low level.

The authors imply in the discussion that the high use of cephalosporins in Chinaa nd Vietnam is because of resistance levels. Prescribing habits and culture may be a stronger driver (such as elevated concerns about allergy in China)

The points about the fragmentation of the market and proliferation of products leading to high transaction costs for government and difficulties for clinicians are important and well made

The authors state that the study does not include drugs purchased in pharmacies, and that there may be significant additional purchases of medicines by patients in primary and secondary care. It is difficult for readers not familiar with the health service utilization patterns to understand how big this contribution might be, and it would be helpful if the authors could offer some idea. Are there studies of service utilization or health expenditures which might provide some evidence??,

Reviewer #2: A well conducted and interesting study looking at the antimicrobial resistance with a focus on pricing.

However the following need to be looked at:

1. The title has to be modified as it does not really relate to patient purchase price as mostly observed in these type of studies.

2. Sentence on line number 6 in the introduction section has to be reformulated as it is not clear. (Overall antibiotic consumption (in DDDs per 1 000 population per day) differs 3 fold between countries .......)

3. In the first paragraph of Study approach: What is the role of provincial health authorities in medicine supply? Why were they purchasing medicines? is it for local use or for which types of facilities?

4. in the section of "Estimation of antimicrobial procurement and the cost of antimicrobials" first paragraph. Is the WHO recommendations similar to Vietnam's standard treatment guidelines? if not, an explanation on the variation with local treatment and why choice of WHO standards should be provided.

5. In the results section, paragraph 1. Does the budget for medication include medical supplies (e.g. gloves, cotton woo, spirit etc)?

6. Table 1, Column 3: what percentage of health facilities was being served by these 52 units included in the study?

7. On the antimicrobial medicines deemed not part of AWaRe, e.g. Nalidixic Acid, are these antibacterials not recommended in Vietnam through standard treatment guidelines?If they are recommended as part of standard treatment guidelines, then an explanation to that effect will be ideal.

8. Table 2 and in the text: List of drugs in each category should be provided as a supplementary table. The reason being that one can not know what is in J01CR, J01EA..etc... If others have to compare with the study the list of those medicines would be ideal

9. Figure 3. The cumulative values should be taken out of it and formulate it into separate graph as there is obscurity of the contributions of primary and secondary health facility in the graph

10. Table 3. For column 2, it will be better to specify how many samples were there per drug not only manufacturer. If the facilities purchased medicine manufactured by the same company but with different prices, it will make sense with various ratios observed in column 4. An example is J02AX04_caspofungin which has ratio of 1.10 even though the manufacturer supplying the medicine to Vietnam is just one.

11. In the discussion, the results in table 3 (column 4) does not relate to comparison with MSH reference price, therefore direct comparison with it as MPR may not be in the best interest. Either, in table 3 recalculate and change the 'High to Low ratio' to Median Price Ratio (MPR) for easy comparison.

12. There is a need for consistency in the use of the following words: 'antibacterials', 'antibiotics' and 'antimicrobials'

Otherwise, the paper is providing a fair way of assessing antimicrobial consumption.

Reviewer #3: This is an original work on costs of antibiotics in Vietnam. Interesting research has been performed, but the current manuscript needs clarity.

GENERAL COMMENTS

English needs to be revised.

Bibliographic references are not accurate or missing. Please, verify their correspondence in the text and the standard required by the journal.

Tables are dense, difficult to understand.

Figures are pixeled.

Pages nor lines are not numbered which complicates the revision.

SPECIFIC COMMENTS

Title

Too large

Introduction

What is the objective of the study?

The definition of the acronym LMIC is mentioned twice.

There is an important focus on the AWaRe classification but the importance of using it to categorize the costs is not clear.

There are confusion when using the terms expenditure, costs and price of drugs. Please harmonize the terms cost/price and specify from which perspective the expenditure is being measured.

Methods

There is substantial need for clarifying the Methods section.

Several concepts are showed in the results/discussion, such as the pharmaceutical market in Vietnam, the different manufacturers/providers of the healthcare system and the relation with neighbour countries.

-Study approach

Please clarify why 52 provincial health authorities were chosen? They represent 2/3 of the data sources and may need different analysis as showed in the graphs.

What are there relation with hospitals?

Do they procure drugs for regional hospitals? Please explain.

What does a health facility represent in Vietnam? Please explain.

Why to choose public instead of private hospitals? The explanation comes too late in the Discussion/Limits.

How were the hospitals chosen? Was it randomly? How did you have access to the “successful tenders”? Normally, these are confidential data and not publicly available. Please clarify.

-Patient and Public Involvement

This section is not necessary, as your data sources were obtained in the “successful tenders”.

Results

Reference 22 is a BMI document published in 2016. Please verify coherence with the text.

Tab 1: Does “other medications” mean all other class of drugs present in the tender ?

Tab 2 is mentioned twice in the text.

Fig 2: what do you mean by “cost shares”? Parenteral formulations and oral formulations have obviously different costs. Why to present a graph? Please explain

Fig 3: what does the cumulative budget represents? Are the drugs bought by the same payer (via different hospitals/provincial health authorities)?

Tab 3: Not readable without the text, please synthesize (3 pages!), and add legend in the end of the table.

By the end of the results there are information regarding manufacturers. This is an interesting focus that should be better explored and previously specified in the Methods section.

Discussion

The description of the use of antibiotics is not clear. Is it about the DDD calculated and indicated in the successful tenders?

Are the facilities chosen representative for the rest of the country? Are they in the same region? Are there social/financial differences in terms of possibility to buy drugs outside the hospital (as mentioned in the text)?

Please check duplicated information in the Introduction and Discussion sections.

Further comparison with southeast countries would be more pertinent than with European countries, specially regarding pharma marketing.

Why IQVIA is mentioned?

Would you suggest why the private sector has lowest prices than the public sector? Hence it would be interesting to explain the reasons to choose to work with the public and not both private and public sectors.

In the Limits section there are several information that should be earlier mentioned, notably in the Methods section.

Conclusion

“Antimicrobials accounted for a third of budget for medication in public hospitals in Vietnam.” This sentence is very ambitious as no representativeness of data analysed was discussed.

The relation between the consumption of “Reserve and non-recommended” antibiotics and the need of a stewardship is not clear.

6. PLOS authors have the option to publish the peer review history of their article (what does this mean?). If published, this will include your full peer review and any attached files.

Reviewer #1: **Yes: **Elizabeth Tayler

Reviewer #2: No

Reviewer #3: No

---

## [Author Response · Author response to Decision Letter 0]

25 Sep 2020

Dear Editor,

On behalf of all authors, I would like to express my sincere appreciation for your careful consideration and your very helpful comments to improve our manuscript. Please find below our response, with contributions from the entire research team, shown in red, for your review and consideration. 

We believe that we have addressed all your suggestions and comments, but please advise us if you consider our manuscript should be further improved. 

Your sincerely

Vu Quoc Dat

Reviewer #1: 

This is an important paper, as there is data on consumption of antibiotics is still relatively scarce, and this is rarely linked to price. The fact that pharmacy and tertiary hospital data means that this is a partial picture, but this paper nevertheless provides a useful contribution to an important issue.

The fact that antibiotics are more than a quarter of the total medicines budget for Vietnam is an important finding. This level of investment is sometimes surprising to policy makers and should encourage them to take antibiotic policies much more seriously. The fact that the proportionate expenditure on reserve antibiotics that are imported, is so high is another policy relevant finding. This study highlights an area where implementation of good public health and prescribing policies could save money and support a shift towards more domestically produced medicines.

The conclusions are slightly surprising. The study has shown significant use and expenditure on Reserve antibiotics and that are not recommended by WHO. The authors conclusion is that there should be price controls on these groups. It is not clear why, as currently financial incentives align with good public health policy. 

The case could be made much more strongly that good stewardship programmes, that might shift consumption towards access groups are likely to result in significant cost savings, and support consumption of domestically produced medicines . This argument would increase the relevance of the study to policy makers and the broader policy debates on antibiotic stewardship, and might stimulate investement in stewardship programmes

[Author's Response To Reviewer Comments] We have revised the abstract and conclusion to explain that there is a large amount of necessary reserve group use in Vietnamese hospitals, due to the high levels of antibiotic resistance in the country. Thus, there is a need to facilitate reserve group use where it is needed but disincentivise reserve use where it is not.

“Antimicrobials accounted for one third of the total spent on medications in study hospitals in Vietnam. The pattern of antibacterial consumption by AWaRe categories was similar to other countries. However, given the relatively high proportion of antimicrobial drug resistance in Vietnam and that although stewardship is important in reducing it, there is necessity for access in certain circumstances to AWARE reserve antibiotics. The optimal approach would be strict stewardship combined with price control, to allow access when needed and prevent access when not needed”.

We also have added information of current antimicrobial resistance in Vietnam in introduction section:

“Vietnam has one of the highest rates of antimicrobial drug resistance in Asia. In an antimicrobial resistance surveillance network of 16 hospitals in Vietnam between 2012 and 2013, the proportion of antimicrobial resistance was high among all pathogens isolated from clinical specimens: penicillin non-susceptible Streptococcus pneumoniae (67%, 229/344 isolates), methicillin-resistant Staphylococcus aureus (MRSA) (69%, 1098/1580 isolates), third-generation cephalosporin-resistant Escherichia coli (56%, 2342/4192 isolates) and Klebsiella pneumoniae (66%, 1479/2227 isolates), carbapenem-resistant Pseudomonas aeruginosa (33%, 578/1765 isolates) and carbapenem-resistant Acinetobacter spp. (70%, 1495/2138 isolates)”.

Most readers won’t be familiar with the Vietnamese health financing system and in particular whether there are financial incentives for practitioners or to hospitals to prescribe antibiotics. It would be helpful if the authors explained whether reimbursement is linked to either the costs of volumes of drugs that are prescribed (as happens in some Asian countries) 

[Author's Response To Reviewer Comments] We have added discussion on the reason for higher prices in public hospitals and commission to prescribers.

“The reasons for this are unclear. In a qualitative study on the price of medication in Vietnam in 2008, the higher prices in public hospitals were suggested to be related to the high bed occupancy rates resulting in a reduced need to attract more patients through competitive pricing, or prices inflated up to 60% by commissions to prescribers and hospital pharmaceutical departments (https://pubmed.ncbi.nlm.nih.gov/28453716/). However, others have shown financial incentives may be not an important factor influencing doctor’s prescribing decision in Vietnam (https://www.ncbi.nlm.nih.gov/pubmed/25793497).”

Inclusion of antifungals is helpful, and the fact that the consumption in an emerging economy (with relatively low HIV) is relatively low is useful information.

Given that the sample did not include many of the tertiary hospitals total consumption of these products may be even higher. Strengthening stewardship programmes to bring prescribing pr

IN the introduction paragraph the authors should note that data quality on consumption patterns is still relatively poor, and so any rankings are questionable .. 

[Author's Response To Reviewer Comments] We have added this note in the introduction session as advised: 

“However, due to lack of resources for collecting reliable data and maintaining surveillance system, data on antibacterial consumption from LMICs are still limited and of poor quality, especially for countries from Southeast Asia”

We were unsure whether the above sentence was complete, it seemed some words were missing in the sentence starting with “Strengthening…” We may therefore not have addressed this comment fully.

The statement that low and middle income countries are disproportionately responsible for the global growth in antibiotics is disingenuous, given that the burden of disease is higher in many of these countries, and their consumption of antibiotics was at a relatively low level.

[Author's Response To Reviewer Comments] It was suggested by Klein’s work (www.pnas.org/cgi/doi/10.1073/pnas.1717295115). We have revised the below sentences for clarity: “Antibacterial consumption was positively correlated with growth in per capita gross domestic product (GDP) and low- and middle-income countries (LMICs) are consequently responsible for driving the rise in global antibacterial consumption.” 

The authors imply in the discussion that the high use of cephalosporins in China and Vietnam is because of resistance levels. Prescribing habits and culture may be a stronger driver (such as elevated concerns about allergy in China)

[Author's Response To Reviewer Comments] We found a systematic review on Irrational Use of Medicines in China and Vietnam (https://doi.org/10.1371/journal.pone.0117710), it shows that the most important factor influencing prescription of irrational medications in Vietnam was lack of knowledge, followed by lack of effective control and regulation mechanisms for drug use whilst it was financial incentives and lack of knowledge in China. 

We have revised and added the text in the Discussion section as below:

“We found that cephalosporins were the most commonly prescribed class of antibiotics in Vietnam. In healthcare sectors in Europe, studies report that the most common antibacterials were beta-lactams, penicillins (J01C). The difference in presecribing patterns may be due to differences in resistance or knowledge. NUMEROUS STUDIES HAVE REPORTED INCREASED RESISTANCE IN VIETNAM COMPRED TO EUORPE [refs] AND whilst in China, where the resistant levels are similar to Vietnam, 3rd-generation cephalosporins were the most consumed antibacterial in hospitals. In a systematic review of studies published between 1993 and 2013 about antimicrobial prescription in China (n=67) and Vietnam (n=29) , the most important factor influencing irrational prescription in Vietnam was lack of knowledge and effective control and regulation mechanisms for drugs use, whilst in China it was financial incentive and lack of knowledge.”

The points about the fragmentation of the market and proliferation of products leading to high transaction costs for government and difficulties for clinicians are important and well made

The authors state that the study does not include drugs purchased in pharmacies, and that there may be significant additional purchases of medicines by patients in primary and secondary care. It is difficult for readers not familiar with the health service utilization patterns to understand how big this contribution might be, and it would be helpful if the authors could offer some idea. Are there studies of service utilization or health expenditures which might provide some evidence??, 

[Author's Response To Reviewer Comments] There are only few studies on service utilization describing this. Whilst there is undoubtedly widespread over-the-counter antimicrobial purchasing in Vietnam, the use of these for hospitalized patients is, from our experience, relatively limited. We have added the below in the discussion:

“We have no data to estimate this, but we consider this proportion to be small. In a prospective study of 892 hospitalised trauma patients admitted to a secondary hospital in Vietnam in 2010, the total medical care out-of-pocket cost paid by patients was US $270.6, 23% of this relating to drugs. However, the out-of-pocket cost for antimicrobials in hospitalised patients was not specified”.

Reviewer #2: 

A well conducted and interesting study looking at the antimicrobial resistance with a focus on pricing.

However the following need to be looked at:

1. The title has to be modified as it does not really relate to patient purchase price as mostly observed in these type of studies. 

[Author's Response To Reviewer Comments] We have revised the title as “Purchase and use of antimicrobials in the hospital sector of Vietnam, a lower middle-income country with an emerging pharmaceuticals market”

2. Sentence on line number 6 in the introduction section has to be reformulated as it is not clear. (Overall antibiotic consumption (in DDDs per 1 000 population per day) differs 3 fold between countries .......)

[Author's Response To Reviewer Comments] We have reworded the text for clarity as follows: 

“The difference in overall antibacterial consumption between the highest and lowest -consuming countries was 3-fold for total use (in DDDs per 1 000 population per day), and up to 16 fold in volume for quinolones and cephalosporins among the mostly high-income countries in the Organisation for Economic Co-operation and Development (OECD)”

3. In the first paragraph of Study approach: What is the role of provincial health authorities in medicine supply? Why were they purchasing medicines? is it for local use or for which types of facilities?

[Author's Response To Reviewer Comments] We have added explanation in the methods section:

“The current medication procurement in Vietnam is mostly implemented through bidding which uses a decentralised (individual hospitals directly conduct the procurement) or centralised model (at national level by ministry of health or at provincial level by provincial departments of health, DoHs). At provincial level, centralised procurement involves provincial DoHs gathering procurement needs of provincial and districts hospitals under their jurisdiction, calling for, reviewing and accepting bids. Hospitals’ estimated requirements for antibacterials are based on consumption in the previous year. Payment is made by the hospitals regardless of whether a decentralised or centralised bid model was used.”

4. in the section of "Estimation of antimicrobial procurement and the cost of antimicrobials" first paragraph. Is the WHO recommendations similar to Vietnam's standard treatment guidelines? if not, an explanation on the variation with local treatment and why choice of WHO standards should be provided.

[Author's Response To Reviewer Comments] We have added the below text in that section: “Currently, there are no national stewardship programmes defining access to different antimicrobials but individual hospitals may have their own policies on their use”.

5. In the results section, paragraph 1. Does the budget for medication include medical supplies (e.g. gloves, cotton woo, spirit etc)?

[Author's Response To Reviewer Comments] No, it does not include medical supplies or medical equipment. We added text for clarity.

“We included tender-winning results totalling US $1.68 billion from 23 secondary hospitals, 7 primary hospitals and 52 provincial departments of health in Vietnam. This excludes disposable and consumable medical supplies and medical equipment”

6. Table 1, Column 3: what percentage of health facilities was being served by these 52 units included in the study?

[Author's Response To Reviewer Comments] We have added explanation of the role of Departments of Health in the methods. Because the DoH gathered bid information from the provincial and district hospitals under their jurisdiction without details of buyers, we can’t provide the exact percentage of health facilities. This was discussed as a limitation in that part of the discussion. However, based on the forecasted estimation of national budget for medication, we estimate that our analysis represents 28.7% of the national budget or medication. 

7. On the antimicrobial medicines deemed not part of AWaRe, e.g. Nalidixic Acid, are these antibacterials not recommended in Vietnam through standard treatment guidelines? If they are recommended as part of standard treatment guidelines, then an explanation to that effect will be ideal.

[Author's Response To Reviewer Comments] We found 4 antibacterials that were purchased and used in Vietnam, but were unclassified by the 2019 AWaRe classification (ticarcillin with a beta-lactamase inhibitor, nalidixic acid, norfloxacin and tinidazole). These are still recommended by national treatment guidelines for specific infections. We have revised these drugs as unclassified throughout the manuscript for clarity and distinguished them from not-recommended drugs by AWaRe.

“We identified 4 antibacterials that were unclassified by the 2019 AWaRe classification (ticarcillin with a beta-lactamase inhibitor (J01CR03), nalidixic acid (J01MB02), norfloxacin and tinidazole (J01RA13) and tinidazole (J01XD02)) but remain recommended by national treatment guidelines for specific infections”

8. Table 2 and in the text: List of drugs in each category should be provided as a supplementary table. The reason being that one can not know what is in J01CR, J01EA..etc... If others have to compare with the study the list of those medicines would be ideal

[Author's Response To Reviewer Comments] We have added a full list of available drugs by category as a supplementary table (Supplementary table 1. List of available antimicrobials) as advised.

9. Figure 3. The cumulative values should be taken out of it and formulate it into separate graph as there is obscurity of the contributions of primary and secondary health facility in the graph

[Author's Response To Reviewer Comments] We would like to keep the cumulative line as it is common in pareto analysis (ABC analysis) which can help to identify antimicrobials that are consuming large parts of the budget. We have added a reference (World Health Organization. (‎2018)‎. Methods to analyse medicine utilization and expenditure to support pharmaceutical policy implementation. World Health Organization. https://apps.who.int/iris/handle/10665/274282.). 

10. Table 3. For column 2, it will be better to specify how many samples were there per drug not only manufacturer. If the facilities purchased medicine manufactured by the same company but with different prices, it will make sense with various ratios observed in column 4. An example is J02AX04_caspofungin which has ratio of 1.10 even though the manufacturer supplying the medicine to Vietnam is just one.

[Author's Response To Reviewer Comments] We have added the number of samples per drug and added the below text under the table:

“Number of samples per drug represents the total number of brands at different strengths procured by all bidders. Some brands may have different strengths and the cost per DDD may vary for the same drug by the same manufacturer. For example, a 10 mL vial of caspofungin contains either 50 mg or 70 mg, therefore the cost per DDD will be different between 2 strengths. Additionally, the same strengths may have different prices in different provinces”.

11. In the discussion, the results in table 3 (column 4) does not relate to comparison with MSH reference price, therefore direct comparison with it as MPR may not be in the best interest. Either, in table 3 recalculate and change the 'High to Low ratio' to Median Price Ratio (MPR) for easy comparison.

[Author's Response To Reviewer Comments] In the MSH reference provided both High/Low ratio and Median Price Ratio (MPR), we intended to examine the variation between the highest unit price with the lowest unit price. In the discussion, we use the H/L ratio to compare the variation of price in our analysis and MSH reference. 

12. There is a need for consistency in the use of the following words: 'antibacterials', 'antibiotics' and 'antimicrobials'

[Author's Response To Reviewer Comments] We have replaced antibiotics by antibacterials. In our manuscript, antimicrobials refers to antibacterials and antifungals 

Otherwise, the paper is providing a fair way of assessing antimicrobial consumption.

Reviewer #3: 

This is an original work on costs of antibiotics in Vietnam. Interesting research has been performed, but the current manuscript needs clarity.

GENERAL COMMENTS

English needs to be revised.

[Author's Response To Reviewer Comments] We have proofread the manuscript. We are happy for the editorial team to make further changes to the English as they see fit.

Bibliographic references are not accurate or missing. Please, verify their correspondence in the text and the standard required by the journal.

[Author's Response To Reviewer Comments] We have updated bibliographies using Plos style.

Tables are dense, difficult to understand.

[Author's Response To Reviewer Comments] We recognised our tables are dense. We will change this table to a supplementary table if the editor agrees. 

Figures are pixeled.

[Author's Response To Reviewer Comments] We have checked the quality of figures to meet the journal requirement. We think these figures were pixelated when they were saved as a pdf file. 

Pages nor lines are not numbered which complicates the revision.

[Author's Response To Reviewer Comments] We have numbered lines as advised. 

SPECIFIC COMMENTS

Title

Too large

[Author's Response To Reviewer Comments] We believe the title – with minor changes - reflects the contents of the manuscript appropriately, also the length is within the length prescribed by the journal

Introduction

What is the objective of the study?

[Author's Response To Reviewer Comments] We have clarified our objective as follows “Our study reports the availability and price of antibacterials and estimates their usage in public hospitals in Vietnam”

The definition of the acronym LMIC is mentioned twice.

[Author's Response To Reviewer Comments] We have removed the duplicate.

There is an important focus on the AWaRe classification but the importance of using it to categorize the costs is not clear.

[Author's Response To Reviewer Comments] We have clarified the importance as follow: 

“Antimicrobials accounted for one third of the total spent on medications in study hospitals in Vietnam. The pattern of antibacterial consumption by AWaRe categories was similar to other countries. However, given the relatively high proportion of antimicrobial drug resistance in Vietnam and that although stewardship is important in reducing it, there is necessity for access in certain circumstances to AWARE reserve antibiotics. The optimal approach would be strict stewardship combined with price control, to allow access when needed and prevent access when not needed”

There are confusion when using the terms expenditure, costs and price of drugs. Please harmonize the terms cost/price and specify from which perspective the expenditure is being measured.

[Author's Response To Reviewer Comments] We have revised the term “prices” for the perspective of the health authorities and hospitals. We also clarified in the Study approach section “Payment is made by the hospitals regardless of whether a decentralised or centralised bid model was used”

Methods

There is substantial need for clarifying the Methods section.

Several concepts are showed in the results/discussion, such as the pharmaceutical market in Vietnam, the different manufacturers/providers of the healthcare system and the relation with neighbour countries.

-Study approach

Please clarify why 52 provincial health authorities were chosen? They represent 2/3 of the data sources and may need different analysis as showed in the graphs.

What are there relation with hospitals?

Do they procure drugs for regional hospitals? Please explain.

What does a health facility represent in Vietnam? Please explain.

[Author's Response To Reviewer Comments] We included 52 provincial departments of health because of the availability of their data. We have added further explanation about this in the Methods section 

“The current medication procurement in Vietnam is mostly implemented through bidding which uses a decentralised (individual hospitals directly conduct the procurement) or centralised model (at national level by ministry of health or at provincial level by provincial departments of health, DoHs). At provincial level, centralised procurement involves provincial DoHs gathering procurement needs of provincial and districts hospitals under their jurisdiction, calling for, reviewing and accepting bids. Hospitals’ estimated requirements for antibacterials are based on consumption in the previous year. Payment is made by the hospitals regardless of whether a decentralised or centralised bid model was used.”

The representativeness of data was indirectly estimated through the national drug expenditure as described in the limitations section in Discussion.

Why to choose public instead of private hospitals? The explanation comes too late in the Discussion/Limits.

[Author's Response To Reviewer Comments] We choose public heath facilities because of the data availability and we recognized it is a limitation of the study. Additionally, the private hospitals accounted for only 5% of total hospital beds. We have updated below information in the study setting. 

“In the private sector, there were 231 private hospitals with 16,000 beds (approximately 5% of national hospital beds) in the country by 2019”

How were the hospitals chosen? Was it randomly? How did you have access to the “successful tenders”? Normally, these are confidential data and not publicly available. Please clarify.

[Author's Response To Reviewer Comments] We have added explanations for the reason to choose hospitals in the Data resources section as advised: “All successful bids with available data were used for this analysis”.

We described in the Methods section that the data was collected from the Drug Administration of Vietnam (DAV) which is the MoH regulatory authority for pharmaceutical products in Vietnam. As requested by the government, this information is public. We have clarified the role of DAV as the MoH regulatory authority.

“Data on the price and characteristics of procured antimicrobials in Vietnam were taken from the successful tenders for medicines in 2018 for hospitals and provincial DoHs in Vietnam as published on the website of the Drug Administration of Vietnam which is the Ministry of Health regulatory authority”

-Patient and Public Involvement

This section is not necessary, as your data sources were obtained in the “successful tenders”.

[Author's Response To Reviewer Comments] We have removed this part.

Results

Reference 22 is a BMI document published in 2016. Please verify coherence with the text.

[Author's Response To Reviewer Comments] We couldn’t find the actual data for the pharmaceutical sales in Vietnam in 2018, therefore we used the forecasted estimation from the BMI documents from 2016.

Tab 1: Does “other medications” mean all other class of drugs present in the tender ?

[Author's Response To Reviewer Comments] Yes, it represents all other classes of drug (non-antimicrobials). We have clarified in table 1. 

Tab 2 is mentioned twice in the text.

[Author's Response To Reviewer Comments] We have removed the duplicate.

Fig 2: what do you mean by “cost shares”? Parenteral formulations and oral formulations have obviously different costs. Why to present a graph? Please explain

[Author's Response To Reviewer Comments] We have changed to “proportion of bidding price”.

Fig 3: what does the cumulative budget represents? Are the drugs bought by the same payer (via different hospitals/provincial health authorities)?

[Author's Response To Reviewer Comments] Cumulative budget represents cumulative percentage of budget for antimicrobials as shown by the secondary horizontal axis on the right. 

Tab 3: Not readable without the text, please synthesize (3 pages!), and add legend in the end of the table.

[Author's Response To Reviewer Comments] We will change this table to a supplementary table if the editor agrees. We have added table legends and abbreviations.

By the end of the results there are information regarding manufacturers. This is an interesting focus that should be better explored and previously specified in the Methods section.

[Author's Response To Reviewer Comments] We have added the below text in the “Materials and Methods”, under “Data resources” sections. 

“We described the antimicrobial manufacturers by their country of origin to estimate the market shares between domestic and international manufacturers which may provide some insight into manufacturing capacity”

Discussion

The description of the use of antibiotics is not clear. Is it about the DDD calculated and indicated in the successful tenders?

[Author's Response To Reviewer Comments] We have used DDD from successful tenders as a proxy for use.

Are the facilities chosen representative for the rest of the country? Are they in the same region? Are there social/financial differences in terms of possibility to buy drugs outside the hospital (as mentioned in the text)?

[Author's Response To Reviewer Comments] We have added a supplementary figure for the data origins, we believe these are representative for the entire country. 

Supplementary figure 1. Mapping of data origins.

Please check duplicated information in the Introduction and Discussion sections.

[Author's Response To Reviewer Comments] We have removed the duplicated paragraph in the Introduction.

Further comparison with southeast countries would be more pertinent than with European countries, specially regarding pharma marketing.

[Author's Response To Reviewer Comments] We recognise the comparison with Southeast Asia and other countries with the same income level is important but there is very limited data from these regions. 

Why IQVIA is mentioned?

[Author's Response To Reviewer Comments] It was mentioned as a source of data used in the WHO report. We tried to provide readers with a context of data sources and we declared to have no conflict of interest. 

Would you suggest why the private sector has lowest prices than the public sector? Hence it would be interesting to explain the reasons to choose to work with the public and not both private and public sectors.

[Author's Response To Reviewer Comments] We chose the public sectors for our analysis due to the availability of data. We have added information to clarify the sampling and data collection. The fact that medication has lower price in the private sectors was observed in previous studies (Nguyen AT, Knight R, Mant A, Cao QM, Auton M. Medicine prices, availability, and affordability in Vietnam. Southern Med Review. 2009;2(2).). 

“The reasons for this are unclear. In a qualitative study on the price of medication in Vietnam in 2008, the higher prices in public hospitals were suggested to be related to the high bed occupancy rates, and lack of motivation to attract more patients through competitive pricing or prices inflated up to 60% by commissions to prescribers and hospital pharmaceutical departments (https://pubmed.ncbi.nlm.nih.gov/28453716/) . However, others have shown financial incentives may be not an important factor influencing doctor’s prescribing decision in Vietnam (https://www.ncbi.nlm.nih.gov/pubmed/25793497).”

In the Limits section there are several information that should be earlier mentioned, notably in the Methods section.

Conclusion

“Antimicrobials accounted for a third of budget for medication in public hospitals in Vietnam.” This sentence is very ambitious as no representativeness of data analysed was discussed.

The relation between the consumption of “Reserve and non-recommended” antibiotics and the need of a stewardship is not clear.

[Author's Response To Reviewer Comments] We have revised the conclusion and removed the relation with antimicrobial stewardship. We hope the representativeness of data has been sufficiently clarified in the revised manuscript.

“Antimicrobials accounted for one third of the total spent on medications in study hospitals in Vietnam. The pattern of antibacterial consumption by AWaRe categories was similar to other countries. However, given the relatively high proportion of antimicrobial drug resistance in Vietnam and that although stewardship is important in reducing it, there is necessity for access in certain circumstances to AWARE reserve antibiotics. The optimal approach would be strict stewardship combined with price control, to allow access when needed and prevent access when not needed.”

---

## [Decision Letter · Decision Letter 1]

5 Oct 2020

Purchase and use of antimicrobials in the hospital sector of Vietnam, a lower middle-income country with an emerging pharmaceuticals market

PONE-D-20-18659R1

Dear Dr. Dat,

We’re pleased to inform you that your manuscript has been judged scientifically suitable for publication and will be formally accepted for publication once it meets all outstanding technical requirements.

Kind regards,

Khin Thet Wai, MBBS, MPH, MA (Population & Family Planning Resear

Academic Editor

PLOS ONE

Additional Editor Comments (optional):

Reviewers' comments:

Reviewer's Responses to Questions

**Comments to the Author**

1. If the authors have adequately addressed your comments raised in a previous round of review and you feel that this manuscript is now acceptable for publication, you may indicate that here to bypass the “Comments to the Author” section, enter your conflict of interest statement in the “Confidential to Editor” section, and submit your "Accept" recommendation.

Reviewer #1: All comments have been addressed

2. Is the manuscript technically sound, and do the data support the conclusions?

Reviewer #1: Yes

3. Has the statistical analysis been performed appropriately and rigorously? 

Reviewer #1: I Don't Know

4. Have the authors made all data underlying the findings in their manuscript fully available?

Reviewer #1: Yes

5. Is the manuscript presented in an intelligible fashion and written in standard English?

Reviewer #1: Yes

6. Review Comments to the Author

Reviewer #1: (No Response)

7. PLOS authors have the option to publish the peer review history of their article (what does this mean?). If published, this will include your full peer review and any attached files.

Reviewer #1: **Yes: **Elizabeth Tayler

---

## [Editor Report · Acceptance letter]

9 Oct 2020

PONE-D-20-18659R1 

Purchase and use of antimicrobials in the hospital sector of Vietnam, a lower middle-income country with an emerging pharmaceuticals market 

Dear Dr. Dat:

I'm pleased to inform you that your manuscript has been deemed suitable for publication in PLOS ONE. Congratulations! Your manuscript is now with our production department. 

Kind regards, 

on behalf of

Dr. Khin Thet Wai 

Academic Editor

PLOS ONE